# Multiphysics Interaction Analysis of the Therapeutic Effects of the Sigmoid Sinus Wall Reconstruction in Patients with Venous Pulsatile Tinnitus

**DOI:** 10.3390/bioengineering10060715

**Published:** 2023-06-12

**Authors:** Zhenxia Mu, Lihui Zhuang, Pengfei Zhao, Bin Gao, Youjun Liu, Zhenchang Wang, Shifeng Yang, Ximing Wang

**Affiliations:** 1Department of Radiology, Shandong Provincial Hospital Affiliated to Shandong First Medical University, Jinan 250021, China; muzhenxia901@163.com (Z.M.);; 2Department of Radiology, Beijing Friendship Hospital, Capital Medical University, Beijing 100050, Chinacjr.wzhch@vip.163.com (Z.W.); 3Faculty of Environment and Life Science, Beijing University of Technology, Beijing 100124, China

**Keywords:** pulsatile tinnitus, sigmoid sinus wall dehiscence, numerical simulation, multiphysics interaction, biomechanical

## Abstract

Sigmoid sinus wall dehiscence (SSWD) is an important etiology of venous pulsatile tinnitus (VPT) and is treated by sigmoid sinus wall reconstruction (SSWR). This study aimed to investigate the therapeutic effects of the different degrees of SSWR and the prognostic effect in patients with VPT. Personalized models of three patients with SSWD (control), 3/4SSWD, 1/2SSWD, 1/4SSWD, and 0SSWD were reconstructed. A multiphysics interaction approach was applied to elucidate the biomechanical and acoustic changes. Results revealed that after SSWR, the average pressure of venous vessel on the SSWD region reduced by 33.70 ± 12.53%, 35.86 ± 12.39%, and 39.70 ± 12.45% (mean ± SD) in three patients with 3/4SSWD, 1/2SSWD, and 1/4SSWD. The maximum displacement of the SSWR region reduced by 25.91 ± 30.20%, 37.20 ± 31.47%, 52.60 ± 34.66%, and 79.35 ± 18.13% (mean ± SD) in three patients with 3/4SSWD, 1/2SSWD, 1/4SSWD, and 0SSWD, with a magnitude approximately 10^−3^ times that of the venous vessel in the SSWD region. The sound pressure level at the tympanum reduced by 23.72 ± 1.91%, 31.03 ± 14.40%, 45.62 ± 19.11%, and 128.46 ± 15.46% (mean ± SD). The SSWR region was still loaded with high stress in comparison to the surrounding region. The SSWR region of the temporal bone effectively shielded the high wall pressure and blocked the transmission of venous vessel vibration to the inner ear. Patients with inadequate SSWR still had residual VPT symptoms despite the remission of VPT symptoms. Complete SSWR could completely solve VPT issues. High-stress distribution of the SSWR region may be the cause of the recurrence of VPT symptoms.

## 1. Introduction

Pulsatile tinnitus (PT) is the auditory perception of rhythmic sounds synchronized with the heartbeat. There are three types of PT: venous PT (VPT), arterial PT, and undetermined [1]. VPT is the most common category of PT, accounting for approximately 84% of patients with PT [2]. Prolonged VPT symptoms are detrimental to the mental health of patients, frequently causing sleeplessness, emotional irritability, anxiety, depression, etc., and are significantly related to critical diseases, such as cerebral hemorrhage, stroke, and intracranial hypertension [3].

Sigmoid sinus (SS) wall dehiscence (SSWD) is a primary cause of VPT, which accounts for about 86.4% of total VPT [4]. SSWD refers to the local bone dehiscence of the temporal bone around the SS vessel. There is no bone tissue between the SS vessel and the temporal bone air cell (TBAC). The temporal bone around the SS vessel acts as a sound barrier for the venous blood flow. Patients with SSWD have a decreased shielding effect on the venous blood flow sound, resulting in venous blood sound transmission to the tympanum [5]. The tympanum, as the auditory receptor of the middle ear, transmits the venous blood flow sound to the inner ear. Eventually, the VPT is generated.

According to the clinical literature, the SSWD region is primarily localized on the anterior lateral of the transverse sinus (TS)-SS junction [6], involving the upper or middle segment of the SS vessel and presenting as a long strip (65.9%) [7]. Patients with SSWD are frequently afflicted with SS diverticulum (SSD), TS stenosis, and high-riding jugular bulbs. The principal therapy approach is SS wall reconstruction (SSWR) surgery. To reduce or resect the diverticulum and reconstruct the SSWD region around the SS, a bone window of 1.5 × 1.0 cm^2^ is drilled centered on the posterior inferior of the TS–SS junction [8]. However, during the postoperative follow-up, up to 31%–57% of patients have inadequate SSWR, different degrees of temporal bone re-cracking, or loss of repair materials [9,10]. The influence of the therapeutic effects of the different degrees of SSWR and the prognostic effect after reconstruction in patients with VPT is unclear, making clinical treatment problematic.

Previous research demonstrated that the place where the blood flow impinges on venous vessels in the TS–SS junction coincides with the SSWD region [6]. We proposed that the impingement of the blood flow on the venous vessel promotes an increase in the wall pressure, which may be correlated to the production of SSWD [6]. Scholars have revealed a long-term impingement of the high wall pressure pulse loading on the temporal bone around the venous vessel, which may result in temporal bone erosion [11]. The density and embrittlement of the temporal bone would deteriorate as time passed, eventually ending in thinning or dehiscence [11,12]. Reducing the impingement of the blood flow on the temporal bone would promote bone hyperplasia and enhance bone density [13,14]. However, there has been no biomechanical and acoustic investigation of the temporal bone surrounding the TS–SS junction after SSWD and SSWR.

The purpose of this study was to investigate the therapeutic effects of the different degrees of SSWR and the prognostic effect after reconstruction in patients with VPT. The transient multiphysics interaction technique was used for revealing the biomechanical and acoustic changes in patients with SSWD and after different degrees of SSWR.

## 2. Materials and Methods

### 2.1. Patients’ Data

Computed tomography angiography (CTA) images of three patients with VPT were acquired. The longest longitudinal diameter of the SSWD region was more than twice the diameter of the SSWR longitudinal window in all three patients. The Institutional Review Board approved the collection of CTA images. Patients’ informed consent was acquired. The patients were all diagnosed with pulse-synchronous and unilateral VPT. The identified pathological characteristics of SSWD, SSD, and TS stenosis were determined to be the etiologies of VPT. A 256-slice spiral CT scanner (Brilliance, Philips Healthcare; Revolution, GE Healthcare, IL, USA) was used to capture the CTA images. The following parameters were used: FOV, 22 cm × 22 cm; detector collimation, 256 × 0.625 mm; matrix, 512 × 512 pixels; slice thickness, 0.625 mm. The patients’ information is displayed in Table 1.

### 2.2. Multiphysics Interaction Geometry Models

The multiphysics interaction geometry models of three patients were reconstructed using the software Mimics 20.0 (Materialise, Belgium), containing the venous blood model, the venous vessel model, the temporal bone model, and the TBAC model (Figure 1a–d). The primary components of the venous model were the inlet, TS, SS, and outlet. The SSD region was the extended zone of the TS–SS junction along the normal curve of the TS vessel towards the normal curve zone of the SS vessel [6]. The SSWD region was defined as the absence of temporal bone tissue between the venous vessel and the TBAC on two or more consecutive cross-section images. Table 2 displays the morphological parameters of geometry models, including the inlet area, outlet area, SSWD area, the longest horizontal diameter of the SSWD region (horizontal axis image), the longest longitudinal diameter of the SSWD region (coronal images), TBAC volume, and tympanum area.

The SSWD region was reconstructed using the software Freeform 17.0 (Freeform, Cary, NC, USA) to simulate the SSWR surgery. On the basis of our previous research, we measured the temporal bone thickness of 20 patients with SSWD near the TBAC zone [6]. The mean temporal bone thickness (1.86 mm) of 20 patients was used to reconstruct the temporal bone in the SSWD region. The longest longitudinal diameter of the SSWD region was divided into four segments to simulate the different degrees of SSWR surgery (Figure 1e, patient 1). The SSWR surgery was carried out by first restoring the SSD region and then reconstructing the temporal bone in the dehiscence region. The current status of the patients was named as the control subject. The cases were labeled 3/4SSWD, 1/2SSWD, 1/4SSWD, and 0SSWD, respectively, based on the residual degree of the SSWD region after reconstruction.

### 2.3. Meshing Generation

COMSOL software 5.6 (COMSOL AB, Stockholm, Sweden) was used for generating tetrahedral grids for all the multiphysics interaction models. The appropriate grid number of the fluid–structure interaction and structure–acoustic interaction was determined using the average displacement of the venous vessel in the SSWD region and the average sound pressure level at the tympanum [15]. By adjusting the size of the grid elements, the number of different grid elements was obtained, and the displacement and sound pressure level were computed. Under different grid element numbers, a relative inaccuracy of less than 5% was deemed acceptable. The maximum grid element size for fluid–structure and structure–acoustic interaction was less than 0.3 mm. The grid numbers of the venous blood, venous vessel, temporal bone, and TBAC were approximately 0.99, 0.19, 0.23, and 0.26 million, respectively. The flow pattern around the venous vessel wall was precisely identified by producing two boundary layers.

### 2.4. Multiphysics Interaction Governing Equations

The ALE formulation of the Navier–Stokes equation was applied in the venous blood fluid component, which replaced the convection velocity with the relative velocity of the moving mesh. Equations (1) and (2) show the momentum conservation and continuity equations [16,17]:(1)$$\rho_{f}\frac{\partial u_{f}}{\partial t}+\rho_{f}\left(u_{f}\cdot \nabla \right)u_{f}=\nabla \cdot \left[-p_{f}I+\mu \left(\nabla u_{f}+{\left(\nabla u_{f}\right)}^{T}\right)\right]+F_{f}$$
(2)$$\nabla \cdot u_{f}=0$$
where $\rho_{f}$ is the venous blood density, $p_{f}$ the venous blood pressure, and μ is the viscosity. The $u_{f}$ represents the venous blood velocity vector, and $F_{f}$ denotes the volume force vector. The I is the unit tensor.

The solid structure components of the venous vessel and temporal bone were governed by Newton’s second law, as described in Equation (3) [17]:(3)$$\rho_{s}\frac{\partial^{2}u_{s}}{\partial t^{2}}=\nabla \cdot \sigma_{s}+F_{s}$$
where $\rho_{s}$ is the solid structure density and $u_{s}$ the velocity vector of the solid structure. The $\sigma_{s}$ is the stress tensor of the solid structure.

The fundamental condition of the fluid–structure interaction followed the dynamic condition and kinematic condition, as illustrated in Equations (4) and (5) [17]:(4)$$\tau_{f}\cdot n=\tau_{s}\cdot n$$
(5)$$d_{f}=d_{s}$$
where $\tau_{f}$ and $\tau_{s}$ are the fluid and solid stress, respectively. Where $d_{f}$ and $d_{s}$ are the displacement of the fluid and solid, respectively. The n is the normal vector of the interface.

In the structure-acoustic interaction research, the acoustic component of the TBAC was governed by the wave equation, ignoring the air viscosity and heat exchange, as displayed in Equation (6) [5]:(6)$$\frac{1}{c_{0}^{2}}\frac{\partial^{2}p}{\partial t^{2}}-\nabla^{2}p=0$$
where $c_{0}$ is the acoustic velocity of the adiabatic air, and p is the sound pressure.

The COMSOL software was adopted to solve the multiphysics interaction governing equations. The fluid–structure interaction question was solved with the fully coupled method. MUMPS direct solver with a maximum iteration number of 50 was chosen to solve the equation. The automatic time step was adopted to accomplish the demands of CFL < 1. The structure–acoustic interaction question was solved using the one-way coupled method. MUMPS suggested a direct solve with a maximum iteration number of 25 was chosen to solve the equation. The convergence precision of the fluid–structure interaction and the structure–acoustic interaction was set to 10^−3^.

### 2.5. Boundary Condition and Calculation Setting

A pulsatile mass flow condition of a patient with VPT was applied at the inlet boundary, while a 0 Pa pressure condition was applied at the outlet boundary [5].

In the fluid–structure interaction research, the maximum Reynolds number was less than 2300 [15,18]. Thus, the venous blood was taken to be laminar, isotropic, homogeneous, and an incompressible Newtonian fluid [18]. The venous vessel and temporal bone were considered to be isotropic and linear [5]. Table 3 shows the material properties of the venous blood, venous vessel, and temporal bone [19,20,21]. The external surface of the venous vessel was in contact with the endocranium, whereas its inside surface was in contact with the temporal bone. The inlet, outlet, and external surface of the venous vessel were all fixed because they were connected to rigid osseous tissue [5]. The contact was defined using the penalty function. The heart cycle time was fixed at 0.8 s. Four cardiac cycles were computed to produce consistent results, and the fourth cardiac cycle was chosen for the fluid–structure interaction research.

The venous vessel wall in the SSWD region was coupled with the TBAC. Venous blood sound was considered to be transmitted only through the air [22]. The displacement of the venous vessel in the SSWD region was used as the input boundary condition for the structure–acoustic interaction research. The TBAC’s acoustic density was set to 1.139 kg/m^3^. The speed was set to 340 m/s [5]. The boundary of the temporal bone was described as an acoustic impedance surface with an impedance of 5.57 MPa∙s/m [23]. The impedance model of the human physiological tympanum, as given in Equation (7) [23,24], was applied to the tympanum.
(7)$$-n\left[-\frac{1}{\rho }\left(\nabla p_{t}-q_{d}\right)\right]=\frac{1}{Z_{i}}\frac{\partial p_{t}}{\partial t}$$
where ρ is the air density. The $p_{t}$ denotes the total pressure. The $q_{d}$ represents the domain volumetric force. The $Z_{i}$ denotes the specific acoustic input impedance of the external domain.

## 3. Results

### 3.1. Blood Flow Velocity

Figure 2 depicts the velocity vector field at the maximum velocity moment of each patient before and after SSWR. In the control subject, the high-velocity blood flow from the TS suddenly turned towards the SS in the TS–SS junction. A portion of the blood flow swirled to the top of the diverticulum, where the vortex was generated. Another component of the blood flow corroded the SS vessel wall on the contact side of the temporal bone as it flowed down the curve of the venous vessel. After SSWR, the blood flow in the TS–SS junction was less turbulent. However, the blood velocity remained high on the side in contact with the temporal bone.

### 3.2. Blood Wall Pressure

Using patient 1 as an example, Figure 3a describes the wall pressure distribution at the maximum velocity moment under different degrees of SSWR. Figure 3b displays the wall pressure distribution at the maximum velocity moment of each patient before and after SSWR. The control subject had a higher wall pressure distribution in the SSD region than the surrounding region. The high wall pressure was primarily distributed in the region where the blood flow impinged on the venous vessel wall. After SSWR, the distribution of the high wall pressure decreased. Despite this, the pressure distribution in the TS–SS junction was still higher than that in the surrounding region.

Figure 4a depicts the average pressure on the SSWD region at the maximum velocity moment for each patient with different degrees of SSWR. The average pressure of patient 1 with 3/4SSWD, 1/2SSWD, and 1/4SSWD was reduced by 30.19%, 30.73%, and 31.44%, respectively, as compared to the control subject (patient 1 329.96 Pa, patient 2 193.83 Pa, and patient 3 95.39 Pa). The average pressure of patient 2 was lowered by 23.29%, 26.86%, and 33.63%, respectively. The average pressure of patient 3 was reduced by 47.61%, 50.00%, and 54.02%, respectively. The average pressure of three patients with 3/4SSWD, 1/2SSWD, and 1/4SSWD was reduced by 33.70 ± 12.53%, 35.86 ± 12.39%, and 39.70 ± 12.45% (mean ± SD), as displayed in Table 4.

### 3.3. Average Displacement of the Venous Vessel

To evaluate the vibration of the venous vessel in contact with the TBAC, the average displacement was calculated. Figure 4b depicts the average displacement of the venous vessel on the SSWD region at the maximum velocity moment of each patient with different degrees of SSWR. The average displacement of patient 1 with 3/4SSWD, 1/2SSWD, and 1/4SSWD was decreased by 40.71%, 52.78%, and 74.30%, respectively, as compared to the control subject (patient 1 17.67 μm, patient 2 6.26 μm, and patient 3 4.01 μm). The average displacement of patient 2 was reduced by 15.18%, 16.36%, and 31.18%, respectively. The average displacement of patient 3 was lowered by 51.51%, 55.85%, and 68.88%, respectively. According to Table 4, the average displacement of three patients with 3/4SSWD, 1/2SSWD, and 1/4SSWD was decreased by 35.8 ± 18.66%, 41.66 ± 21.97%, and 58.12 ± 23.49% (mean ± SD), respectively.

### 3.4. Von Mises Stress of the Temporal Bone

Figure 5 and Figure 6 describe the von Mises stress distribution of the temporal bone at the maximum velocity moment for each patient with different degrees of SSWR. The high von Mises stress was distributed at the SSWD region edge and the temporal bone in contact with the SSD region in the control subject (Figure 6). The high von Mises stress of the reconstruction region was lowered after SSWR. Especially after complete reconstruction of the SSWD region, the von Mises stress of the temporal bone in contact with the TS–SS junction vessel was higher than that in the surrounding region. The von Mises stress distribution characteristics of the temporal bone were similar in all three patients.

### 3.5. Displacement Distribution of the Temporal Bone

Figure 7 depicts the displacement distribution of the temporal bone at the maximum velocity moment for each patient in the TS–SS junction before and after SSWR. High displacement was distributed at the bottom edge of the SSWD region and the temporal bone in contact with the SSD swelling region in the control subject. After SSWR, the high displacement of the reconstruction region was reduced. However, the displacement of the temporal bone in contact with the TS–SS junction vessel was higher than in the surrounding region. Furthermore, the position of the high displacement distribution was correlated to the position of the high von Mises stress distribution.

To estimate the displacement difference of the SSWR region, the maximum displacement of the SSWR region at the maximum velocity moment for each patient was computed, as displayed in Figure 4c. As shown in Table 4, the maximum displacement of three patients with 3/4SSWD, 1/2SSWD, 1/4SSWD, and 0SSWD was decreased by 25.91 ± 30.20%, 37.20 ± 31.47%, 52.60 ± 34.66%, and 79.35 ± 18.13% (mean ± SD), respectively.

### 3.6. Sound Pressure Level

Figure 4d depicts the sound pressure level at the tympanum of each patient with the first mode frequency. The maximum sound pressure level was recorded for each patient with different degrees of SSWR at 1200 Hz, 2000 Hz, and 1000 Hz, which matched the first model frequency at the tympanum. Compared to the control subject (patient 1 65.16 dB, patient 2 73.53 dB, and patient 3 66.79 dB), the sound pressure level of patient 1 with 3/4SSWD, 1/2SSWD, 1/4SSWD, and 0SSWD was reduced by 23.91%, 30.09%, 52.49%, and 124.97%, respectively. The sound pressure level of patient 2 was reduced by 21.73%, 17.12%, 24.03%, and 115.04%, respectively. The sound pressure level of patient 3 was reduced by 25.53%, 45.87%, 60.34%, and 145.36%, respectively. The sound pressure level of three patients with 3/4SSWD, 1/2SSWD, 1/4SSWD, and 0SSWD was decreased by 23.72 ± 1.91%, 31.03 ± 14.40%, 45.62 ± 19.11%, and 128.46 ± 15.46% (mean ± SD), respectively, as shown in Table 4.

## 4. Discussion

Previous research primarily concentrated on the morphology of venous vessels and the pathogenesis of VPT. In this study, from the perspective of VPT treatment, the multiphysics interaction approach was chosen to combine biomechanics and acoustics to explore the therapeutic effects of the different degrees of SSWR and the prognostic effect after reconstruction in patients with VPT. Along the longest longitudinal diameter of the SSWD region, multiphysics interaction models of three patients with SSWD, 3/4SSWD, 1/2SSWD, 1/4SSWD, and 0SSWD were reconstructed. The transient multiphysics interaction approach was adopted to explore the changes in biomechanics and acoustics under different degrees of SSWR. The research results revealed that the pathway of the blood flow in the TS–SS junction was altered after SSWR, which improved the impingement of the blood flow on the venous vessel, lowered the blood wall pressure, and prevented the transmission of the blood vessel vibration to the inner ear, thereby solving VPT. The degree of VPT may be attenuated in the condition of partial SSWR, but there was still a residual VPT issue. The VPT issue could be eliminated after completing SSWR. The high-stress distribution in the TS–SS junction of the temporal bone after SSWR may be the cause of postoperative recurrence of VPT symptoms.

The pulsating venous blood impinges the venous vessel on the SSWD region, causing the venous vessel to vibrate [5,18]. The vibration is then conveyed from the SSWD region along the TBAC to the inner ear and auditory nerves, inducing the production of VPT [15]. The research results found that during the generation process of VPT, the blood flow pattern in the TS–SS junction was changed after SSWR, which improved the impingement of the high velocity on the venous vessel (Figure 2). Thus, the wall pressure and vessel vibration in the TS–SS junction were reduced (Figure 3 and Figure 4b). During the conduction process of VPT, the displacement of the venous vessel in the SSWD region was more than 10^3^ times that of the temporal bone in the SSWR region (Figure 4b,c). As a consequence, the SSWR region had a shielding effect, preventing vessel vibration from reaching the tympanic. Under these two influences, the sound pressure level was lowered during the formation and transmission processes of the venous blood sound.

In the short term after SSWR, the CTA characteristics of some patients with VPT were associated with residual SSWD [10], particularly in patients whose SSWD region involved the upper or middle segments of the SS. The patients displayed various degrees of remission in VPT [25]. The findings revealed that after 1/4 degree of SSWR (3/4SSWD), blood flow impingement on the venous vessel in the TS–SS junction was considerably improved, and the wall pressure was lowered by approximately 33.70 ± 12.53%. The reconstructed temporal bone effectively shielded the high wall pressure in the TS–SS junction (3/4SSWD). Previous research discovered that the vibration of the venous vessel was primarily driven by wall pressure [15]. The displacement of the temporal bone was far less in magnitude than the venous vessel displacement in the SSWD region, which was reduced by approximately 25.91 ± 30.20%. The sound pressure level was reduced by approximately 23.72 ± 1.91% (control subject 68.49 ± 4.44 dB). Affected frequencies with a threshold recovery of more than 15 dB were regarded to be an effective cure in clinical evaluation [26]. This explained the reason why patients with inadequate SSWR still have residual VPT symptoms despite VPT symptoms remission. Especially after 3/4 degree of SSWR (1/4SSWD), the sound pressure level was reduced by 45.62 ± 19.11%. Depending on the classification level of hearing loss, patients with this degree of VPT remission generally recovered [26]. A complete SSWR could completely solve VPT issues (0SSWD).

In the long term after SSWR, the mending materials were partially fused (48.5%) or separated (15.2%) from the normal temporal bone structure in some patients [10]. The patient was required to undergo revision surgery following surgical reconstruction [25]. Because of the inherent morphology of the venous vessel, the TS–SS junction region remained laden with high pressure after SSWR (Figure 3b). The high-pressure distribution was consistent with the high-stress distribution of the reconstructed temporal bone region (Figure 6). The vibration of the reconstructed temporal bone region was primarily driven by stress (Figure 7). The prolonged vibration loading on the reconstructed temporal bone region may cause the partial fusion or complete separation of the mending materials from the normal temporal bone structure. Particularly for patients with TS stenosis, the impingement of the blood flow on the temporal bone at the distal of the TS stenosis became stronger [11,14,27]. Therefore, firmer materials, such as bone cement or bone chips, may be better than an autogenous bone meal in terms of minimizing the recurrence of SSWR [25,28,29].

The velocity pattern and wall pressure distribution of the venous vessel in this study were consistent with the findings of previous findings in similar venous geometry model conditions [5,30]. Furthermore, in vitro experiments in our previous study validated the correctness of the fluid–structure interaction simulation [15]. Kim discovered that patients with VPT had a frequency range of 125–2000 Hz, with heard tinnitus pulsating signals around 1000 Hz [31]. Tian reported that the sound pressure level at the tympanic of a patient with VPT in a first mode frequency was 56.9 dB [5]. The first mode frequency of the three patients with VPT in this study was in the low frequency range, and the sound pressure level was comparable to earlier studies. In general, compared with the previous research results, the results of this study closely matched the general research conclusions.

The current study had several limitations. First, the sample size of this study was insufficient since only patients with the SSWD region involving the upper and middle segments of the SS were evaluated. Second, the glue-fixed gap between the autogenous bone and normal temporal bone structure was neglected in the simulation of SSWR surgery. Third, the reconstructed SSWD region was assumed to be a linear elastic material, which was consistent with the material property of the normal temporal bone region. In the actual SSWR surgery, the autogenous bone meal was reconstructed to fill the SSWD region, and the fascia was placed over the bone slices. The reconstructed SSWD region material properties are quite complex. Finally, given the lack of specific patient inlet boundary velocity data, the boundary condition from the literature was applied in the multiphysics interaction analysis. Future work will focus on patient cohort studies, obtaining the patient personalized blood flow velocity data by 4D flow magnetic resonance imaging, and obtaining the material properties of repair materials through in vitro experiments to further improve the accuracy of the study.

## 5. Conclusions

The pattern of blood flow in the TS–SS junction was changed after SSWR, and the pressure of the venous vessel was lowered. The reconstructed temporal bone effectively shielded the high wall pressure in the TS–SS junction and prevented the blood vessel wall vibration from reaching the inner ear. Despite the remission of VPT symptoms, several individuals with inadequate SSWR experienced residual VPT symptoms. A complete SSWR could eliminate the VPT issues. The high stress of the reconstructed temporal bone region after SSWR may be the cause of postoperative recurrence of VPT symptoms. The completion of this work provided a theoretical foundation and suggestions for personalized early warning and effective treatment of patients with VPT.

## Figures and Tables

**Figure 1 bioengineering-10-00715-f001:**
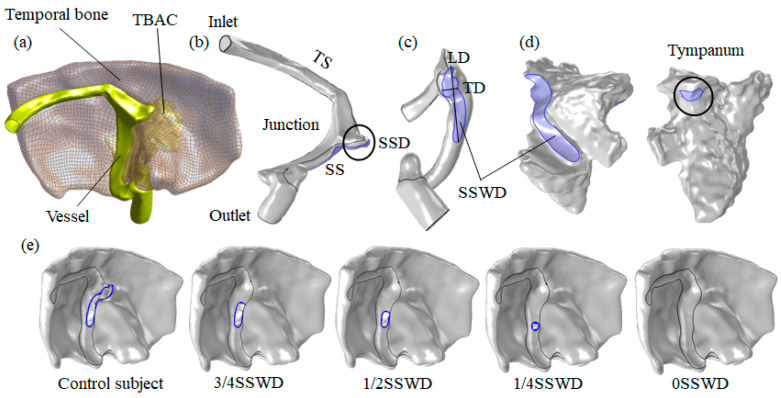
3D multiphysics interaction geometry reconstruction. Using patient 1 as an example: (**a**) Position of the venous blood vessel, temporal bone, and TBAC. (**b**) Position of the inlet, TS, SSD, SS, and outlet. (**c**) Position of the SSWD longest transverse diameter (TD), SSWD longest longitudinal diameter (LD), and SSWD region on the venous vessel. (**d**) Position of the SSWD region and tympanum on TBAC region. (**e**) Different degrees of SSWR.

**Figure 2 bioengineering-10-00715-f002:**
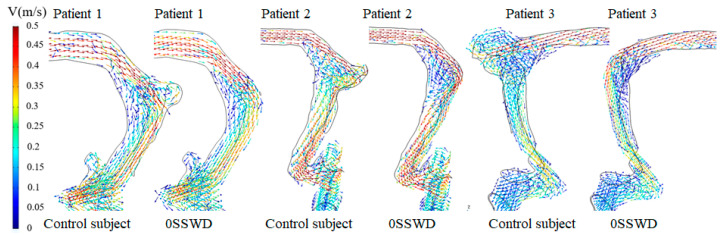
Blood velocity vector at the maximum velocity moment of each patient before and after SSWR.

**Figure 3 bioengineering-10-00715-f003:**
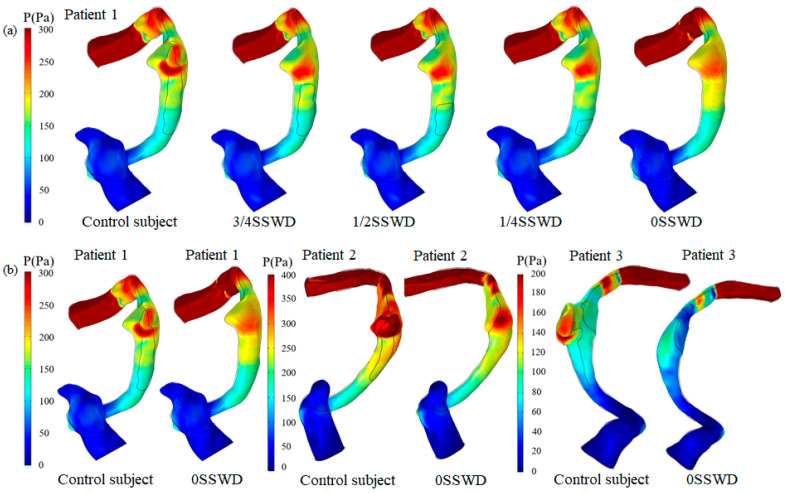
(**a**) Pressure distribution at the maximum velocity moment of patient 1 with different degrees of SSWR. (**b**) Pressure distribution at the maximum velocity moment of each patient before and after SSWR.

**Figure 4 bioengineering-10-00715-f004:**
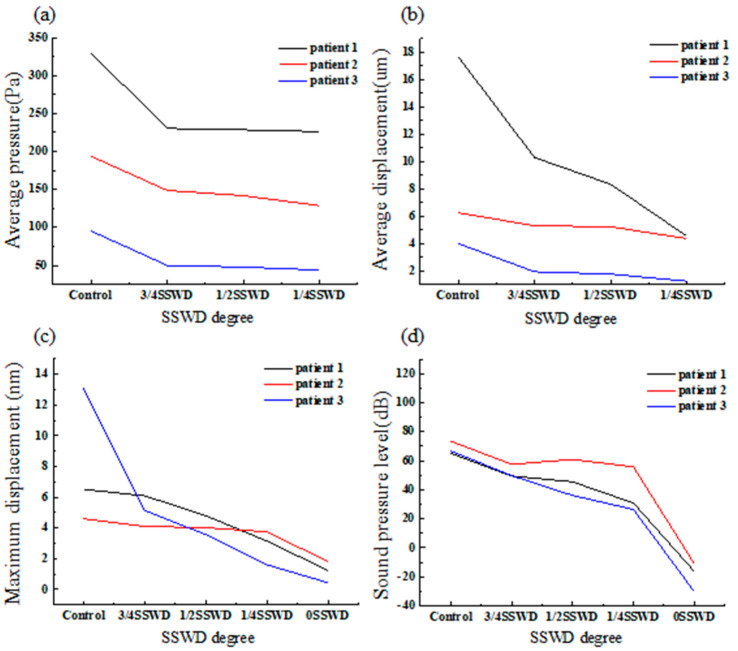
(**a**) Average pressure on SSWD region at the maximum velocity moment for each patient with different degrees of SSWR. (**b**) Average displacement of the venous vessel wall on SSWD region at the maximum velocity moment for each patient with different degrees of SSWR. (**c**) Maximum displacement of SSWR region at the maximum velocity moment for each patient with different degrees of SSWR. (**d**) Average sound pressure level at the tympanum for each patient with different degrees of SSWR.

**Figure 5 bioengineering-10-00715-f005:**
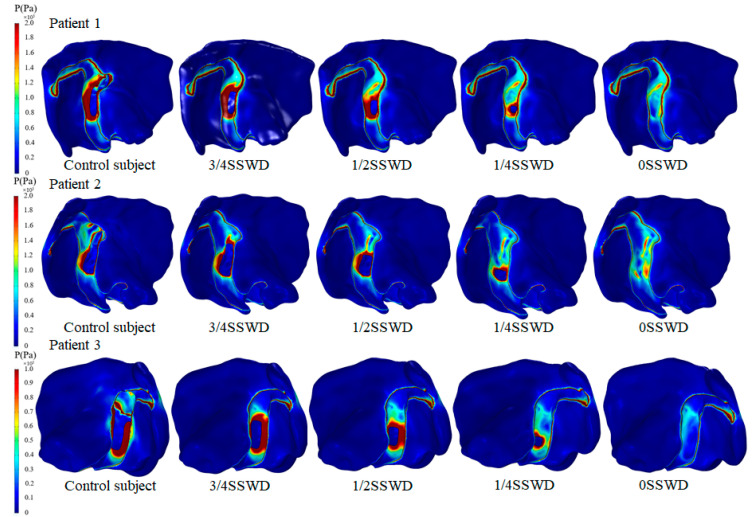
Von Mises stress distribution of the temporal bone at the maximum velocity moment for each patient with different degrees of SSWR.

**Figure 6 bioengineering-10-00715-f006:**
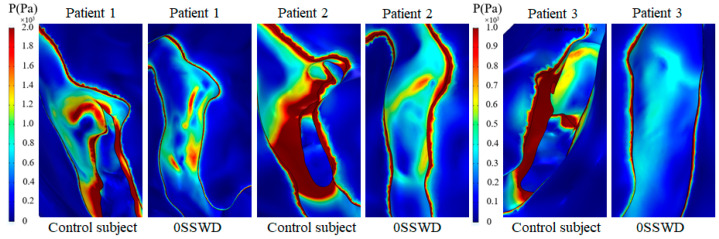
Von Mises stress distribution of the temporal bone at the maximum velocity moment for each patient at the TS–SS junction before and after SSWR.

**Figure 7 bioengineering-10-00715-f007:**
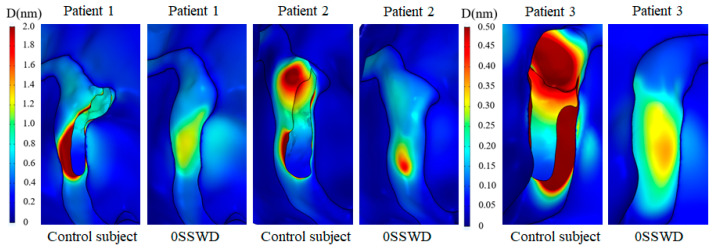
Displacement distribution of the temporal bone at the maximum velocity moment for each patient in the TS–SS junction before and after SSWR.

**Table 1 bioengineering-10-00715-t001:** Patients’ information.

Patients	Age	Gender	Duration (Years)	Side	Diagnosis
patient 1	38	female	6	left	SSWD + SSD + HJB
patient 2	41	female	4	left	SSWD + SSD + HJB
patient 3	55	female	5	right	SSWD + SSD

Sigmoid sinus wall dehiscence (SSWD); Sigmoid sinus diverticulum (SSD); High jugular bulb (HJB).

**Table 2 bioengineering-10-00715-t002:** Geometry morphological parameters.

Patients	Inlet Area (mm^2^)	Outlet Area (mm^2^)	SSWD Area (mm^2^)	Longest Horizontal Diameter of SSWD (mm)	Longest Longitudinal Diameter of SSWD (mm)	TBAC Volume (mm^3^)	Tympanum Area (mm^2^)
patient 1	31.95	63.26	131.68	5.04	27.46	6834.90	18.56
patient 2	40.83	60.73	159.45	5.49	28.46	5814.80	16.14
patient 3	22.80	56.78	179.72	7.86	30.35	8763.30	16.31

**Table 3 bioengineering-10-00715-t003:** Material properties.

Name	Density (g/cm^3^)	Viscosity (Pa∙s)	Elastic Module (MP)	Poisson’s Ratio	Reference
Venous blood	1.05	0.0035	-	-	Mu et al. [18]
Venous vessel	1.05	-	1.26	0.3	Cho et al. [19] and Lofink et al. [20]
Temporal bone	2.00	-	12,000	0.3	Du et al. [21]

**Table 4 bioengineering-10-00715-t004:** The biomechanical and acoustic parameters decreased with different degrees of SSWR (mean ± SD).

Parameters	Control Subject	3/4SSWD	1/2SSWD	1/4SSWD	0SSWD
Average wall pressure (Pa)	206.39 ± 117.79	33.70 ± 12.53%	35.86 ± 12.39%	39.70 ± 12.45%	-
Average displacement (μm)	9.31 ± 7.32	35.8 ± 18.66%	41.66 ± 21.97%	58.12 ± 23.49%	-
Maximum displacement (nm)	8.06 ± 4.44	25.91 ± 30.20%	37.20 ± 31.47%	52.60 ± 34.66%	79.35 ± 18.13%
Sound pressure level (dB)	68.49 ± 4.44	23.72 ± 1.91%	31.03 ± 14.40%	45.62 ± 19.11%	128.46 ± 15.46%

Average wall pressure of the venous blood on the SSWD region; Average displacement of the venous vessel on the SSWD region; Maximum displacement of the temporal bone on the SSWR region; Sound pressure level of the tympanum.

## Data Availability

The original data are available from the corresponding author upon an appropriate request.

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
