# Peer review of "Multiphysics Interaction Analysis of the Therapeutic Effects of the Sigmoid Sinus Wall Reconstruction in Patients with Venous Pulsatile Tinnitus"

_bioengineering, 2023, doi:10.3390/bioengineering10060715_

Round 1
Reviewer 1 Report
This manuscript is a computational study of sigmoid sinus wall dehiscence, responsible for most venous pulsatile tinnitus cases, and its surgical therapy, the sigmoid sinus wall reconstruction. The study is information-rich and nicely illustrated. Nevertheless, its impact would increase by eliminating the following weak points:
1. The Introduction needs to be revised to clarify the objectives of the present study. A single sentence (lines 74-75) is not enough to explain them. The methodological details from lines 75-80 would fit in Section 2, whereas the conclusion from lines 81-83 should be postponed for Section 5.
2. Statements concerning material properties need to be backed up by literature citations. In Subsection 2.5, the choice of model parameters should be explained in more detail. Please include an extra column in Table 3 with the corresponding literature citations. Also, on line 380, the authors mention that in their model the temporal bone was considered a linear elastic material, and this assumption was consistent with the material property of the temporal bone. A reference is needed here, and an explanation of why is it considered a limitation of the study, even though it is consistent with the literature. Also, what are the options for going beyond that limitation in future studies?
3. The text contains many acronyms, which affect its legibility. The authors are encouraged to use them sparingly, for terms that appear several times in the text. For example, TSS is not explained (I only assume it stands for TS stenosis) and only appears once in the text (line 91). For the necessary acronyms, I would include list of abbreviations to help the reader to find them right away.
Minor comments:
a. Line 124: Please remove the periods from figure references. For example, instead of "Figure. 1e", simply write "Figure 1e". Proceed in the same way for all subsequent figures mentioned in the main text.
b. Lines 148, 150, 153, 154, 157, 158, and 162: Please do not start a sentence with a mathematical symbol. Also, explain symbols more specifically. A statement such as "I is the second-order tensor" is not informative because many types of second order tensors are used in physics.
c. Line 391: Instead of "recurrence in patients" I would write "recurrence of VPT symptoms".
Reviewer 2 Report
The presented research on Pulsating Tinnitus aimed at clarifying via numerical simulations the effects of Wall Reconstruction. It was professionally conducted and the obtained results were reported in a satisfactory way.
I spotted a number of points that need the intervention of the authors to fix (small) mistakes and to improve the clarity of presentation. Please see the marked-up copy of the paper in attachment. Also, I have included a comment (on the last page) that came to mind while reading Section 3.6.

The literal presentation is easy to read and follow, a few minor mistakes need to be fixed.
